# Current Practices and Potential Nanotechnology Perspectives for Pain Related to Cystic Fibrosis

**DOI:** 10.3390/jcm8071023

**Published:** 2019-07-12

**Authors:** Laura M. Trandafir, Magdalena M. Leon, Otilia Frasinariu, Ginel Baciu, Gianina Dodi, Elena Cojocaru

**Affiliations:** 1Pediatric Department, Grigore T. Popa University of Medicine and Pharmacy of Iasi, 700115 Iasi, Romania; 2Medical I Department, Grigore T. Popa University of Medicine and Pharmacy of Iasi, 700115 Iasi, Romania; 3Pediatric Department, “Dunărea de Jos” University of Galati, 800008 Galati, Romania; 4Advanced Centre for Research-Development in Experimental Medicine, Grigore T. Popa University of Medicine and Pharmacy of Iasi, 700115 Iasi, Romania; 5Morpho-Functional Sciences Department, Grigore T. Popa University of Medicine and Pharmacy of Iasi, 700115 Iasi, Romania

**Keywords:** pain, cystic fibrosis, children, adults, nanotechnology, quality of life

## Abstract

Pain is a complex, multidimensional process that negatively affects physical and mental functioning, clinical outcomes, quality of life, and productivity for cystic fibrosis (CF) patients. CF is an inherited multi-system disease that requires a complete approach in order to evaluate, monitor and treat patients. The landscape in CF care has changed significantly, with currently more adult patients than children worldwide. Despite the great advances in supportive care and in our understanding regarding its pathophysiology, there are still numerous aspects of CF pain that are not fully explained. This review aims to provide a critical overview of CF pain research that focuses on pain assessment, prevalence, characteristics, clinical association and the impact of pain in children and adults, along with innovative nanotechnology perspectives for CF management. Specifically, the paper evaluates the pain symptoms associated with CF and examines the relationship between pain symptoms and disease severity. The particularities of gastrointestinal, abdominal, musculoskeletal, pulmonary and chest pain, as well as pain associated with medical procedures are investigated in patients with CF. Disease-related pain is common for patients with CF, suggesting that pain assessment should be a routine part of their clinical care. A summary of the use of nanotechnology in CF and CF-related pain is also given. Further research is clearly needed to better understand the sources of pain and how to improve patients’ quality of life.

## 1. Introduction

Cystic fibrosis (CF) is a complex, progressive, life-limiting genetic disorder that affects many systems of the human body, namely the respiratory, digestive and reproductive systems, all of which contribute to impaired health-related quality of life [1]. CF is caused by mutations in the cystic fibrosis transmembrane conductance regulator (CFTR) gene protein, typically found in the apical membrane of most epithelial tissues including lungs, submucosal, salivary, gall bladder glands, gut, kidney, pancreas, cervix and uterus [2]. Rarely, CFTR is also expressed in non-epithelial cells and tissues, including ventricular cardiomyocytes and aortic smooth muscle cells, neurones in the brain, corneal and vascular endothelial cells and lymphocytes [3].

Located on chromosome 7 in the region of 7q31.2, CFTR predominantly functions as a chloride channel, which affects sodium and water movement across the cell membrane. Since the discovery of the defective CF gene in 1989, 2065 mutations have been listed in the Cystic Fibrosis Mutation Database (CFTR1) [4], grouped into six distinct classes according to CFTR synthesis, structure and impact at a cellular level, namely: no protein production, defective protein processing, defective channel regulation, decreased conductance, reduced functional CFTR protein and diminished stability [5,6].

According to the World Health Association (WHO) and available CF registers, there are approximately 80,000 CF patients worldwide, with the prevalence varying according to ethnicity [7]. The WHO gene diseases global prevalence map suggests that in Europe, one in 2000–3000 newborns is found to be affected by CF, with discrepancies between countries, while in USA the incidence is reported to be one in every 3500 births. On the contrary, the existing evidence indicates that CF prevalence in Africa and Asia is rare, due to severe underdiagnosis. Cystic Fibrosis Foundation (CFF) [8] data show that the probability of people with two defective CF genes, one from each parent (named carriers), having a child with CF is 25%, 50% that the child will be a carrier but will not have CF, and 25% that the child will be a non-carrier and will not have CF.

Currently, CF patients’ life expectancy has increased worldwide, as explained in a 35-year observational study in France [9], due to early diagnosis, routine screening and improvements in disease management, adequate nutritional support and new treatments. However, this drop has a drawback: namely, adult CF patients tend to experience additional health challenges (diabetes, osteoporosis, hearing loss, arthropathies, musculoskeletal problems, infertility, etc.) that increase the number of inpatient hospitalizations related to CF, along with an increase in the associated costs by about 138.31% according to Agrawal et al. [10].

In recent years, pain has also been recognized as a potential health challenge in CF, reported to affect 75% of children [11] and 89% of adults [12]. In a comparative study of the prevalence of pain symptoms, characteristics and occurrence in children and adults with CF, pain was significantly more intense and lasted notably longer among adults, but its rate and recurrence did not differ and was not related to the severity of CF [13]. The sources of pain are various: it accompanies pulmonary, digestive and articular manifestations; it appears as a complication of the CF disease; it is secondary to clinical tests (hypopharyngeal aspiration, bronchoalveolar lavage, functional respiratory tests, pleural puncture) or to administrated treatment (postural drainage, respiratory physiotherapy, evacuation enema) [14]. The most prevalent locations were the abdomen for children, and the back, head and chest for adults.

Studies have shown that pain assessment and treatment in CF are imperative, especially as pain has been associated with risk of both exacerbation and death, interferes with activities and reduces the quality of life of chronically ill patients. Moreover, in clinical practice, physicians noticed an increasing number of adult CF patients who confirmed painful symptoms but could not estimate its frequency. In this context, it is essential to provide a comprehensive analysis of pain in this chronic disease.

Therefore, the primary aim of this systematic review was to establish a pain assessment, with prevalence, characteristics and clinical association in children and adults with CF, impact on life quality and future prospects for advanced nanotechnology or other strategies.

Reviewing the CF pain literature in this context indicated that further research exploring the measurement of pain is necessary. The dynamic character of the field is reflected in the increasing number of recent publications. A recent query on the search engine ScienceDirect returned 32,470 hits for the keywords “cystic fibrosis pain”, representing a massive increase from 1996 with 429 publications to 1519 in 2018, 869 in 2019 and 44 in 2020.

## 2. Pain in CF

The International Association for the Study of Pain (IASP) emphasizes that “pain is always subjective” [15]. Self-evaluation, which includes descriptions of pain feelings, images, experiences and information on the quality and intensity of spatial‒temporal dimensions of pain, is regarded as the “gold standard” of measurement [16].

### 2.1. Pain Assessment Tools

Pain, considered a vital signal, is a subjective symptom and consequently cannot be measured by objective methods; therefore, the clinician must depend on the patient to supply key information on the localization, quality and severity of the pain. Pain scores have gained acceptance as the most accurate and reliable way of assessing a patient’s pain and response to pain treatment. There are multiple measures available to assess and monitor pain, of which the most popular is the visual analogue scale (VAS), followed by the verbal rating scale (VRS) and the numerical rating scale (NRS), evaluated as unidimensional procedures [17]. Other available scales used for pain assessment are descriptive, pictorial, and mixed [18]. VAS is a simple, reproducible and effective pain-intensive determination, widely used in the clinic, correlated with pain measurements on verbal and numerical scales. VAS consists of a 10 cm horizontal or vertical line having at the ends the notations: “no pain” at level 0 and “worst pain imaginable” at level 10 [19]. The patient is instructed to put a mark on the line where he feels that his level of painful sensation would be [20]. The distance from the “0” end is visually analogous to his pain, used as a numerical index of pain severity. The advantages of VAS as a measure of pain intensity are related to its reporting property, compared to other measurement methods, the ease of using it and obtaining the score and minimal invasiveness. The disadvantages of VAS are the unidimensional conception of pain since the term “pain” involves different qualities.

As already established in the Introduction, pain is a common problem in patients with CF, although it is not considered a painful disease. Multiple studies published over the past decade describe a high prevalence of severe pain in individuals with CF, but its impact on the clinical outcomes and quality of life of these people is often underestimated. No single pain assessment tool is available to measure all aspects of pain in CF.

Pain assessment typically involves documenting pain location, frequency, intensity, and physiological and psychological/social factors by means of pain assessment tools.

To our knowledge, the first identified pain assessment tool was medical chart review, used to examine the incidence and therapy of chronic pain in a group of older patients with CF [20]. Following Ravilly et al. [20], another pain assessment tool, namely the multidimensional pain inventory, was utilized in a sample of adult outpatients with CF to assess their perception of pain and its psychosocial consequences [21]. In 2011 Kelemen et al. [12] described the intensity and location of pain and its relationship with health-related quality of life and pain catastrophizing in 73 adults with CF using three questionnaires, the Brief Pain Inventory, the Pain Catastrophizing Scale and the CF-Quality of life as pain assessment tools. The results assessed pain intensity as generally mild, suggesting that the pain catastrophizing scale could be used by clinicians and physiotherapists to identify patients who should be targeted for pain management.

In 2013, Havermans et al. [22] provided a description of all the pain assessment tools in CF that authors employed from January 1995 to December 2012. According to the authors’ review, there is a range of assessment tools to measure pain in CF, many of them developed by the researchers/clinicians themselves without validity or reliability considerations, as follows:

- Medical chart review used for pain location and frequency evaluation on CF patients [20,22];

- Multidimensional pain inventory that covers the impact on CF patients’ life quality [22];

- Faces pain scale and revised versions, for pain severity or intensity, frequency, duration and bother [23];

- Body outline drawings [24];

- Eland pain location for children using colour-based intensity principle [23];

- Daily diaries for 28 days [23];

- Ad hoc or web questionnaires/interviews about pain location, intensity and frequency, pain management, chronicity and the impact on life quality and CF treatment management: visual analogue scale [25];

- Pain disability index that assesses how pain disrupts normal activities [26], and

- Brief pain inventory (BPI) (a validated pain tool) and modified versions to ensure CF-specific content [27].

Recent studies highlight the use of the NRS and cystic fibrosis symptom scale (CFSS) as assessment tools for the evaluation of pain.

In 2018, Tomaszek et al. [28] used the NRS tool for pain intensity evaluation in adolescents and young adults with CF. A score of 0 meant no pain, 1–3 was mild, 4–6 was moderate and 7–10 was given for severe pain. The study included 95 Polish patients aged between 14 and 25 years who reported at least one source of pain over the prior two weeks. Mild pain was described by 20% of the subject patients, moderate pain by 45% and severe pain by 35%. The obtained results correlated pain intensity with poor quality of life.

In 2019, Maras et al. [29] assessed pain using a personalized pain subscale (CFSS) developed for this study. This tool is comprised by a 41-item questionnaire that evaluates the frequency of pain signs in the past seven days using a five5-point Likert scale from 0 to 4 (4—all the time) and 17 pain subscales from 0 to 16 (16—the highest number of indicated pain symptoms). The results of this study suggest that pain significantly contributes to a poor quality of life in patients with CF.

In another study, 250 adults with CF were evaluated in a complex longitudinal study that explored the relationships between social support and health outcomes such as mental and physical health, treatment activity and quality of life [30]. Pain, as a physical symptom, was measured using the memorial symptom assessment scale and the physical symptom subscale (MSAS-PHYS). The five-point Likert scale, ranging from 0 to 4, monitored the severity, distress and frequency of symptoms. Even though social support was associated with health outcomes, future research should distinguish the impact of pain alone.

### 2.2. Prevalence and Location of Pain

The variety of pain locations and the wide range in prevalence of CF patients reporting pain are provided in Table 1.

The studies of Koh et al. [14] and Palermo et al. [31] showed that 46% of children with CF reported a painful episode at least once a week, with the intensity associated with common CF-related procedures. The primary reported locations of pain were the abdominal/pelvic region (50%), chest (37%) and head/neck (33%). In contrast, a study by Munck et al. [23] found a very low incidence (6%) of recurrent abdominal pain in children with CF. This low percentage was obtained by using a strict definition of recurrent abdominal pain, namely “at least three bouts of pain, severe enough to affect activities, over a period of not less than three months, with attacks in the year preceding the examination” [36] using Apley’s criteria, which led to exclusion from the study of many children with less severe pain.

Using daily online diaries, Blackwell and Quittner [11] found that 76% of adolescents had episodes of pain most frequently in the stomach (49%), followed by the head/neck (42%) and chest (37%), pain associated with treatment burden, respiratory symptoms, higher levels of depression and anxiety. The study of Lechtzin et al. [32] also demonstrated that pain is common in paediatric CF patients and was associated with increased pulmonary distress and decreased quality of life. Pain was most often encountered in the abdomen (42.0%), head/sinuses (31.9%), joints (18.8%), chest (10.1%), back (5.8%) and muscles (2.9%).

Sermet-Gaudelus et al. [13] reported that 59% of children (73 patients aged 1–18 years) with CF had at least one episode of pain in the previous month. The most prevalent combination in children was headache with abdominal pain, similar to previous studies. Abdominal pain caused by the multitude of abdominal pathologies in CF was attributed to 60% of subjects, followed by 19% for muscular pain, 16% for chest and backache, 10% for stomach, 11% for arthralgia and 3% for cervicalgia. The same study also included 110 adults (18–52 years) who reported pain during the previous month. The two populations did not differ significantly for prevalence or recurrence rates, but the pain was considerably more intense and lasted notably longer in the adult population. The authors quantified the pain locations as follows: backache: 50%, abdomen: 36%, headache: 35%, chest: 26%, arthralgia 20%, cervicalgia: 10%, stomach: 7%, and muscular pain 6%.

In the study conducted by Ravilly et al. [20], out of 78 adult patients with CF and chronic pain, 55% expressed headache (sinusitis, nasal polyps, migraine, arterial hypertension or hypercarbia), 65% chest pain (pleural, muscle, pneumothorax, costal fractures), 19% back pain, 19% abdominal pain and 16% limb pain.

Festini et al. [34] evaluated the prevalence of pain symptoms and the influence of pain symptoms on patients’ lives in 17% of the whole Italian adult CF population by using specific questionnaires. The obtained results indicated a high prevalence of painful episodes among CF adult patients, in terms both intensity and frequency, in a two-month period; 32.6% of patients experienced intense to severe headache, gastric pain and backache.

A web-based education program tailored to adolescents and young adult patients with CF who may be feeling pain reported that 55.6% experienced daily moderate pain episodes lasting up to two hours in the chest (72%) and limbs/joints (61%) [26].

Hayes et al. reported similar results [33] and suggested that pain is usual in patients with CF, with the most common sites being the head (64%), sinuses (54%), back (50%), chest (46%) and abdomen (34%), followed by the knees (30%), wrists (19%) and fingers (5%) for 82% of patients within the past month. A total of 73 participants were included during a period of clinical stability, with 33 repeating the measurements during an acute illness at the Alfred Hospital. Based on the completed questionnaires, mild pain was reported by 89% of stable participants, with high prevalence of back, buttock and hip pain and 79% of those with exacerbations. Six principal areas of pain were identified for clinically stable patients and those with an acute exacerbation, namely neck/head (52%/50%), back/hips/buttocks (70%/31%), chest (20%/35%), and upper (23%/27%) and lower (27%/20%) limb.

In 2015, Michel-Cherqui et al. [35] determined the prevalence of pain in a population of lung transplant candidates as 59%. Among 143 patients who were included in this prospective study, 34% exhibited back, head and joint pain, 28% chest and 19% abdomen and limb pain.

The recent meta-analysis performed by Lee et al. [1] showed that the pooled prevalence of pain in adults with CF was 77% and in children was 42%, and common regions of pain included the back, abdomen, chest and limbs. The authors recommend that validated and age-appropriate questionnaires should be applied across the lifespan to ensure an accurate reflection of pain as experienced by both children and adults.

From the presented data, the most common location of pain is at the thoracic and spinal levels. Although the studies highlighted a large individual variety of pain localization in patients and a wide range of prevalence between studies, the most common location of the pain varied from the abdomen in paediatric studies to the back, head and chest in adolescent and adult populations with CF. Also, it can be observed that the frequency of painful recurrences did not differ significantly in children compared to adults and the pain was not correlated with the severity of CF. The frequency of painful episodes in children varied between one per day and less than once per month, and the duration of painful episodes ranged from less than 30 min to a few days.

### 2.3. Gastrointestinal Indicators and Abdominal Pain

Gastrointestinal manifestations in CF patients, as detailed in Table 2, are either a direct consequence of the CFTR dysfunction or a secondary complication of the disease or therapy [37] in the intestinal epithelium, pancreatic and biliary ducts with viscous secretions and local inflammatory phenomena, all of which cause maldigestion, intestinal transit disorders and intestinal obstruction.

In CF, the digestive tube and the annex glands display prenatal and postnatal changes. Macroscopically, a thick, mucosal adherent meconium appears in the gut in the 16th–17th week, which can cause a thin bowel occlusion known as meconium ileus [45]. Histological sections at the wall level reveal marked hyperplasia of caliciform cells, with the production of abundant mucous secretion along with dilation and viscous secretions in the lumen of oesophageal, Lieberkühn and Brunner duodenal glands, ileocecal appendix and rectum glands.

The pancreatic parenchyma is affected from the 19th–20th week of intrauterine life in 85–90% of cases. Morphologically, the increase of secretory cells of the pancreatic acinus and the enlargement of intra -and extralobular ducts occur, with the accumulation of a homogeneous eosinophilic material in the lumen [46]. In general, changes in pancreatic parenchyma are correlated with the age of the patient and the severity of the disease. Interstitial fibrosis occurs in long-term evolution and therefore the pancreas becomes small, atrophic, with an increased consistency, with hyperplasia of adipose tissue, the presence of cystic structures and intra- and extralobular ducts obliterated by adherent secretions at the surface epithelium level [45]. Hodson et al. [47] have described four degrees of pancreatic damage in optical microscopy that may co-exist in the same patient: the presence of intraductal eosinophilic accumulations; duct dilation associated with exocrine atrophy, small cystic structures, intra- and interlobular fibrosis; atrophy of pancreatic acini and liposclerosis in the parenchyma; and obliteration of ducts with total destruction of acini, destruction of endocrine pancreas and massive fibrosis. The endocrine pancreas is affected by extensive fibrosis, with damage of Langerhans islet vascularization, hypoxia with secondary morphological destruction and replacement with adipose tissue, especially in young diabetic patients.

Prenatal and postnatal changes also affect the liver and bile ducts. At this level, proliferation of bile ducts and chronic inflammation, associated with fibrosis, occurs. Focal bile duct fibrosis is a characteristic feature of CF and is being discussed in cases of prolonged neonatal jaundice [48]. The macroscopic examination reveals an abundant mucoid material in the lumen of the intra- and extrahepatic bile ducts, as well as in the gallbladder’s lumen, where calculi can also appear. Intracellular accumulation of triglycerides with the appearance of hepatic steatosis is another common aspect in CF [45]. In a small percentage, patients developed macronodular biliary cirrhosis, sometimes associated with portal hypertension [49]. Other morphological aspects of the bile duct are represented by sclerosing cholangitis, distal stenosis of the common hepatic duct, obstruction or even atrophy of the cystic duct, gallbladder atrophy or fibrosis or secondary biliary lithiasis [50].

#### Mechanisms of Pain Occurrence

Abdominal pain is classified as visceral, parietal and sympathetic, and the mechanisms by which pain is perceived are multiple and complex.

The visceral pain is triggered by the stretched mechanoreceptors of the smooth muscles in the cavity organs, and is transmitted through non-myelinated C nerve fibres. Distensions, spasm and inflammation trigger the signals that are transmitted to the brain. Clinically, pain has a diffuse, cramp or severe burn nature, hard to locate, and can supplement autonomic responses such as facies erythematosus, sweating and nausea [51].

Somatic pain is caused by profound somatic injuries cutting, breaking, and inflammation of the organs covered by parietal peritoneum. Transmission is fast, via myelinated Aδ fibres, propagated through somatic spinal nerve circuits, so parietal pain tends to be intense and localized.

Sympathetic pain occurs when there are relationships between the sympathetic nervous system and the central nervous system, usually intense, but making the differential diagnosis more difficult. Depending on the organ where the pain comes from, abdominal pain is felt in different topographical areas of the abdomen (Table 3). Also, depending on the stimulus, the pain can be diffused or localized.

Several aetiologies of pain have been identified by specialists and are briefly presented in Table 4. Thus, abdominal pain may be explained by reflux disease, pancreatitis, constipation, intestinal obstruction and hepatic or gallbladder injury (Table 4).

Along with the increase in the survival rate, the aetiology of chronic abdominal pain as well as its pattern of development has changed. Numerous studies conducted in both children and adults have revealed a high incidence of pain through life, regardless of the severity of the disease. An illustrated study made of 73 children and 110 adults with CF revealed that chronic abdominal pain has the highest prevalence in children, at about 60% compared with only 36% in adults; by comparison, the prevalence of abdominal pain in healthy children is approximately 11% [50]. Munck et al. also showed that eight out of 130 children with CF experienced recurrent abdominal pain, with a strong impact on the quality of life [23].

In children, pain secondary to pancreatic insufficiency is dominant and involves abdominal meteorism, chronic diarrhoea and steatorrhea. Against the backdrop of chronic pain, there are episodes of acute abdomen, secondary to distal intestinal obstruction syndrome and intestinal invagination. Patients with distal intestinal obstruction syndrome have crampy abdominal pain in the right lower quadrant, abdominal distension and vomiting in combination with palpable mass, flatulence and weight loss, which produces an obstruction in the ileocecal junction, usually seen on simple abdominal radiography [54]. The risk factors associated with the occurrence of distal intestinal obstruction syndrome are: severe genotype, pancreatic insufficiency, past organ transplants and/or other episodes in a history of the disease [51].

In terms of the intensity of abdominal pain, most children with CF claim abdominal discomfort and just a few true recurrent abdominal pains. These children with severe pain also reported a negative impact on their lives [26].

Painful recurrent pancreatitis episodes are also described in adult CF patients, along with an increase in the prevalence of gastrointestinal cancers [55,56].

### 2.4. Pulmonary Indicators and Chest Pain

In CF, the respiratory system is severely affected in both the upper and lower respiratory tracts and in the pulmonary alveoli, as presented in Table 5.

In the nasal mucosa, hyperplasia of the mucosecretory glands takes place at the corium level, and polypoid structures can often be seen, usually at the base of the ostium of the ethmoidal and maxillary sinuses. Studies performed in the second trimester of pregnancy showed elevated levels of mucin in the tracheobronchial gland [47]. After birth, submucosal gland hypertrophy, obstruction of excretory ducts and intense multiplication of tracheobronchial mucosal cells associated with caliciform cell metaplasia in the respiratory epithelium, even in the absence of an inflammatory process, were observed [46]. The presence of an infectious process is correlated with mucosal ulceration outbreaks and the formation of granular tissue areas at this level, accumulation of chronic inflammatory elements, proliferation of the respiratory epithelium along with the formation of papillary structures and its squamous metaplasia. The most common bacteria involved in pulmonary infections are *Staphylococcus aureus*, which can also produce necrosis outbreaks of parenchyma, *Haemophilus influenza, Pseudomonas aeruginosa* and *Burkholderia cepacia* [45,57]. Histologically, normal bronchiole, caliciform cells are replaced by Clara cells, but in CF they are present and typically hypertrophied.

At the pulmonary alveoli level, the microscopic examination reveals recurrent pneumonia, with the presence of an inflammatory exudate rich in PMN in the lumen, which finally produces progressive destruction of parenchyma [45].

Pulmonary hypertension, characterized by a progressive fibrosis of the vascular walls, with increased intravascular pressure and the occurrence of bleeding and haemoptysis, appears in evolution. The presence of pulmonary hypertension is associated with impairment of respiratory function [58]. Long-term development is accompanied by airway wall fibrosis diffuse bronchiectasis, abdominal outbreaks and alveolar septal destruction with emphysema.

Progressive pulmonary disease is the main cause of morbidity and mortality in CF. CF manifestations are often characterized as presented in Table 5, by chronic cough, purulent sputum and fatigue, all contributing to impairment of health-related quality of life and reduced exercise tolerance [59]. The viscous secretions in the airways favour the chronic respiratory tract infections, possibly with *Pseudomonas* species, usually a mucoid phenotype.

Chest pain in CF is caused by pulmonary damage such as pneumonia, pleurisy, serositis, thromboembolism, pulmonary hypertension, pneumothorax and other causes that are related to the chest wall or musculoskeletal pain.

Pneumothorax can cause a sharp pain that occurs suddenly and is accompanied by shortness of breath. Some researchers maintain that the large mucous plugs caught in the small bronchi may cause pain in CF, even though it is known that the lungs do not have pain-sensing nerves [60].

Pain can be also related to pulmonary function investigations (respiratory functional tests, bronchoalveolar lavage) or therapeutic operations (bronchial drainage, physiotherapy). Sermet-Gaudelus et al. [13] found that 80% of children and 90% of adults reported pain during or after bronchoscopy. Multiple pulmonary exacerbations, requiring repeated antibiotic treatments, have been associated with increased pain intensity. For example, the decrease in forced expiratory volume in 1 s (FEV1) is associated with an increased risk of pulmonary exacerbations and antibiotic associations. Thus, for every 1% increase in FEV 1, it was observed that the risk of pulmonary exacerbation decreased by 4%, thus reducing the risk of pain as well. In adults, the risk of death increased in patients with high scores for pain assessment. Therefore, pain in adult CF patients is associated with an increased risk of exacerbation and death [22]. Also, Koh et al. [14] showed that in children with chest pain the risk of functional limitation of chest excursions and decrease of FEV1 increases.

An increased incidence of chest pain has been reported in patients with advanced pulmonary disease. Chest pain increases significantly three months before death [22]. The reasons for associating pain with unfavourable clinical development have been discussed in the literature and are represented by a decrease in airway mechanical clearance because the patients either limit their movements to combat pain or barely tolerate the therapeutic operations that exacerbate their pain, which leads to a vicious circle of worsening infections and increased risk of new exacerbations. On the other hand, pain can be considered a clinical marker of exacerbation of inflammation and of “disease activity,” so it can be an indicator of the progressive, fast and severe evolution of CF. Therefore, correct assessment of pain can help establish the therapeutic protocol, which in this case requires complex, aggressive therapy to prevent unfavourable development and the appearance of new complications, as well as to improve the quality of life of these patients [33].

Perception of pain, not only as a clinical symptom but also as a factor affecting optimal treatment, including physiotherapeutic techniques, should be a priority in the daily practice of any caregiver. This is highlighted by a study that found that pain during physiotherapeutic treatment increased the risk of future exacerbations. Physiotherapy is an important element of CF therapy, so pain control contributes to a proper conduct of this manoeuvre, and thus combats future exacerbations [33].

### 2.5. Musculoskeletal Pain

Regarding bone, chest and back pain, orthopaedic disorders are described in the literature as possible aetiologies. Osteoporosis is a complication of CF and can cause fractures with characteristic pain. Changes in posture due to kyphosis, reduced muscle tone and paravertebral muscle contractions are other causes of pain present in CF patients [62].

Patients with CF may have manifestations of monoarthritis or polyarthritis and pulmonary hypertrophic osteoarthropathy. CF arthropathy occurs especially in knee and ankle joints and is characterized by recurrent pain-associated swelling episodes, lasting up to 5–7 days [51]. The symptomatology is absent between these acute pain episodes.

Bamberger‒Marie osteoarthropathy or hypertrophic pulmonary osteoarthropathy has a low prevalence of 5% in patients with CF [63]. The clinical trial is characterized by the presence of digital clubbing, arthralgias and ossifying periostitis and is associated with cutaneous manifestations [64].

In a study of 105 adults with CF with a mean age of 30 years, the majority of patients (90%) reported musculoskeletal chronic pain for at least three months and only 10% reported acute pain of less than three days’ duration. The use of combined therapies of manual mobilization and massage techniques improved breathing and reduced pain [24].

### 2.6. Pain Associated with Medical Procedures

The severity of this chronic illness requires many repeated medical operations, which are reported as painful by patients.

It is important to distinguish between the general and procedural pain reported by both children and adults during common CF-related procedures in all studies. Sermet-Gaudelus et al. [13] reported that approximately 85% of CF children and 78% of CF adults described at least one episode of procedural pain in the previous month; these episodes were up to 1.5 times more frequent in patients with severe disease. While 28% of CF children and 10% of CF adults reported pain during chest physiotherapy, nearly 80% of children and 83% of adults experienced pain during capillary blood sampling [13]. In contrast, Koh et al. [9] noted mild and unfrequented procedural pain in the majority of children. Thus, only a small number of children experienced important pain from peripherally inserted central catheter placements (9%), throat cultures (8%) and blood sampling (7%).

Sometimes pain in patients with CF may appear as a side effect of the chronic administration of medication. Thus, prolonged antibiotic therapy in repeated respiratory infections causes intestinal dysbiosis, which maintains inflammation in the intestine, intestinal transit changes, intestinal permeability changes and maldigestion, clinically translated into chronic diarrhoea, abdominal meteorism and abdominal pain. Chronic replacement therapy with pancreatic enzymes can cause fibrotic colorectal cancer with obstruction of intestinal transit accompanied by abdominal pain [65].

## 3. Inflammation and Pain Aspects

A defect in CFTR protein leads to impaired secretion of chloride and an increase in sodium absorption, which causes airway surface liquid depletion, defective mucociliary clearance and reduced mucus clearance. The secretory cell products have a low water content, adhere to the epithelium and are difficult to remove, therefore encouraging bacterial colonization, recurrent infections, chronic inflammation and irreversible damage to the airway epithelium [66,67]. The vicious cycle of inflammation and infection causes the alteration of various organs such as the lungs, pancreas, liver, and intestines.

CF lung disease is characterized by early colonization and infection of the airways associated with chronic inflammation, which leads to permanent structural damage to the CF airways and impaired lung function [68]. Cantin et al. [68] illustrated some of the potential links between CFTR deficiency, namely innate and acquired immunity dysregulation, cell membrane lipid abnormalities, various transcription factor signalling defects, as well as altered kinase and toll-like receptor response and intense airway inflammation, linking the CF basic defect to inflammation. The regulatory mechanisms of CF airway inflammation are not fully understood, but epithelial dysfunction plays a vital role in the development of specific inflammatory responses. The neutrophils (or polymorphonuclear leukocytes, PMNs) are central to the disease process in the CF airway being present in massive quantities due to a combination of increased influx and decreased clearance. Even if neutrophils represent the first cells migrating into the pulmonary compartment to combat bacterial and fungal pathogens, their role in CF airway pathophysiology is harmful through the release of oxidants and proteases (elastase) that interfere with CFTR expression and/or function [69]. Even if the *CFTR* gene implication is unclear, neutrophil elastase stimulates the airway inflammatory response by delivering serine proteases and matrix metalloproteases that cause a variety of harmful effects, and correlate with lung function deterioration and respiratory exacerbations [70].

The CF patient airway includes numerous pro-inflammatory mediators in different concentrations, including TNF-α (tumour necrosis factor- α), IL-1β (interleukin), IL-6, IL-8, IL-17, IL-33, GM-CSF (granulocyte-macrophage colony-stimulating factor), G-CSF (granulocyte-colony stimulating factor), and HMGB-1 (high-mobility group protein 1) [71,72]. While TNF-α stimulates neutrophil oxidative activity and secretory responses associated with cachexia, IL-1β encourages acute phase responses, fever and muscle protein catabolism [73]. GM-CSF has many pro-inflammatory purposes, but the most relevant to CF is represented by its ability to delay neutrophil apoptosis and inhibit their clearance from the lungs. Since the CF airways are in a state of chronic inflammation, these pro-inflammatory mediators only maintain this vicious circle. In the illustrated simplified view of the CF airway inflammatory response, Nichols et al. highlighted the CF airways’ deficiency in several counter-regulatory molecules including IL-10, nitric oxide (NO), and lipoxin-A4 (LXA4) [74]. The IL-10 molecule ends the acute inflammatory response, decreases the chemokines production, hinders pro-inflammatory transcription factors and induces neutrophil apoptosis. Inflammation could also be promoted by decreased NO synthesis that intensifies abnormal airway surface liquid height and impairs the ability of airway smooth muscle to relax [75]. The LXA4 lipid mediator stimulates the resolution of inflammation and reduces bronchoalveolar damage [76].

In a study from 2006 on 40 patients with chronic widespread pain, the authors found low levels of the anti-inflammatory cytokines IL-4 and IL-10. These findings may indicate a crucial role in the pathophysiology of chronic prevalent pain, a hypothesis that may eventually help in supporting the diagnosis of pain and guiding the appropriate treatment approach [77]. Another study from 2007 recognizes the anti-inflammatory cytokines’ physiological role in inflammation and pathological role in systemic inflammatory states, with IL-10 cytokine having the most potent anti-inflammatory properties [78].

Intestinal inflammation in CF has been extensively studied in recent years [79]. Although a defective CFTR is the primary promoter and the exact underlying mechanism is unknown, there have been various mechanisms proposed: intestinal dysbiosis (due to both intense intestinal secretions and prolonged antibiotic therapy for respiratory diseases), inappropriate secretion of digestive enzymes (maldigestion), intestinal transit disorders and intestinal resections (e.g., intestinal ileus) [80].

According to Smyth et al. [81], the inflammatory biomarkers measurements on 21 children with pancreatic insufficient CF and 12 controls demonstrated increased values of albumin, IgG, IgM, interleukin- (IL-) 1β, IL-8, neutrophil, elastase, and eosinophilic cationic protein, suggesting that there is immune activation in the gastrointestinal mucosa of children with CF, which may result from the basic cellular defect. In a separate study that compared duodenal mucosal specimens from 14 pancreatic insufficiency CF patients, 20 healthy controls, and four non-CF patients with chronic pancreatitis, an increased mononuclear cell infiltrate expressing the intercellular adhesion molecule (ICAM -1), the IL-2 receptor α (CD25), IL-2 and interferon- γ has been observed at the lamina propria level of the duodenal mucosa. These results also suggest that the perturbation of local mucosal immune response may contribute to the overall clinical picture in CF patients [82].

The incidence of intestinal inflammation in children with CF was assessed by Bruzzese et al. [83] in 30 pancreatic insufficiency CF children compared to 30 healthy controls using faecal calprotectin and rectal NO production. The obtained data suggested that intestinal inflammation is a virtually constant feature in patients with CF, and faecal calprotectin concentration and rectal NO production are two non-invasive reliable markers of intestinal inflammation.

## 4. Pain and CF Patients’ Quality of Life

According to the literature data, clinicians and specialist studies/case reports, acute, recurrent or chronic pain interferes with daily life and is related to poor quality of life regardless of the age of CF patients. It has been assumed that the location of pain affects health-related quality of life differently, as chest pain is more commonly associated with poor mood [31,84].

The first study to examine pain in relation to adherence, psychological symptoms, and health-related quality of life in adolescents with CF using an online diary was performed in 2015 [11]. The results confirmed that pain is common among adolescents with CF and negatively affects their health-related quality of life (e.g., sleep, daily mood fluctuations or treatments).

According to Sermet-Gaudelus et al. [13], 50% of children and 70% of adults reported that pain intensity and recurrence significantly altered their quality of life, by restricting physical activity for 28% of children and 44% of adults, or causing asthenia for 27% and 58%, respectively. The patients also reported that pain caused absenteeism from school or work (15%) and negatively affected their family life (27%).

Although pain appears to be less severe in the paediatric population than in adults, it still has a negative impact on daily activities, emotional, physical and social functioning, body image, eating, treatment burden, health-related quality of life, pulmonary and digestion health, as indicated by Lechtzin et al. [32].

For 59.8% of subjects, painful symptoms are the main cause of unfavourable effects on their life [34]. The negative consequences of pain on daily life reported by patients were mostly related to restrictions in carrying out normal everyday activities, along with sleeping disorders and lost days at work, school or studying. According to Festini et al. [34], the assessment of pain should become a routine procedure in CF clinical examination and attention to pain-related problems should become an integral part of care in CF centres.

Over the 12-month period, pain frequently interfered with general activities for 41.9%, mood for 56.8%, and work for 47.3% of CF adults at Johns Hopkins Hospital. Symptoms of depression and anxiety, as well as lower quality-of-life scores, were also associated with the presence of pain.

More importantly, pain interfered with regular CF treatment, being related to the ability or inability to perform daily care procedures [14,26,31,33]. The most important restriction related to pain in CF patients is linked to chest physiotherapy. Hayes et al. [33] reported a decreased compliance with mechanical airway clearance, because CF treatment exacerbates the pain. So, the refusal to perform systematic chest physiotherapy is associated with a high risk of chronic infection exacerbation or lung colonization with new germs. Other risk factors for disease progression are anxiety and depression due to pain, which is associated with an unfavourable prognosis for these patients, as studies associated depression with poor clinical outcome in patients with chronic diseases [1,37].

Although pain is a frequent manifestation in CF, of all patients who experienced pain syndrome, less than half asked for CF team advice and about a quarter of them discussed with their family or friends [34].

## 5. Pain and CF Management

### 5.1. CF Treatment Perspectives

Maintenance of adequate treatment status is essential to CF care. In this sense, an attractive perspective on pain management that must be taken into account is efficient CF treatment, since pain is a potential complication of the disease.

Currently, CF treatment, which includes antibiotics, anti-inflammatory drugs, bronchodilators, mucolytics and osmotic agents, aims to attenuate disease progression and delay the beginning of irreversible lung damage, but unfortunately has complications in terms of daily regimens [85].

Recent advances in the treatment of CF have focused on correcting the defective CFTR protein. The new Ivacaftor personalized or genomically guided therapy for CF, a potent CFTR modulator in patients carrying G551D mutation, has been shown to improve gastrointestinal symptomatology and relieve pain. According to Accurso et al. [86], around 5% of the total CF population with CF is eligible for treatment with ivacaftor worldwide.

According to Fajac and De Boeck [87], there are currently several new potentiators being evaluated in clinical trials, such as QBW251 from Novartis (phase II), GLPG1837 from Galapagos (phase II), or CTP-656 from Concert Pharmaceuticals (phase I), but their relatively high costs will limit access in lower-income countries.

Other pharmacological strategies under investigation involve the development of various agents that:

- increase the amount of CFTR, namely the amplifier PTI-428 from Proteostasis Therapeutics [87];

- stabilize CFTR at the cell membrane, such as the stabilizer cavosonstat from Nivalis that inhibits the S-nitrosoglutathione reductase [88]; this stabilizer is currently under investigation in Phase II in combination with Orkambi® and Kalydeco®; 

- over-read the premature stop codons: Ataluren agent, tested in a new phase III trial in patients with CF bearing a nonsense mutation who do not receive inhaled tobramycin [87]; and

- bypass the CFTR channel by identification of alternative chloride channels that should facilitate the development of specific activators [89].

### 5.2. Nanotechnology Perspectives on CF

A recent review argues that a multidisciplinary approach comprised of physicians, psychologists, pharmacists, chemists, material science specialists, engineers and therapists is required to address the optimal care of CF patient [90].

According to Leso et al. [91], nanotechnology, one of the most innovative tools of the century, will transform the biomedical field by improving prevention, diagnosis and disease therapy. Indirectly, the nano-personalized strategy facilitates a better quality of life and life expectancy for patients with CF.

Based on specific small size, large surface area-to-volume ratios, improved pharmacokinetics and minimal side effects, nano-sized materials have shown potential as carriers for targeted drug delivery in pharmacotherapy [92]. An important advantage of drug delivery-based nanotechnology is the improved diffusion and degradation characteristics of the encapsulated nanomaterial, allowing the drug to be protected during its transit to the target, while maintaining the biological and chemical properties and allowing it to be released at an appropriate and effective rate.

Therefore, the main question among scientists and clinicians is whether nanotechnology will bring new hope for CF patients dealing with pain.

In the specific area of CF treatment, nanomedicine occupies an important position in the identification, design and development of new molecular targets. Gene therapy and nano-selective and sustained drug delivery systems can improve drug therapies systemically and locally [93].

The success of CF inhalation therapies, according to d’Angelo et al. [85], is related to the penetration of the anatomical/biological barriers imposed by the CF lung such as drug deposition and lung distribution, CF mucus, cellular barriers (bacteria biofilm and airway epithelial cells) and phagocytosis. In designing new inhalable drug delivery strategies for CF patients, researchers must take into account several mandatory parameters, namely engineered size, density and shape for lung deposition, drug incorporation, delivery and macrophage clearance escape.

Table 6 presents the drug-loaded delivery engineered systems under development in the last decade for overcoming the “biobarriers” enforced by the CF lung.

The most commonly employed nonviral nanosized carriers are made of cationic lipids and polymers [85]. Lipid-based carriers, namely liposomes and solid lipid nanoparticles for drug delivery and cationic lipid particles for gene delivery, are mainly composed of phospholipids and cholesterol. These advanced lipid formulations present remarkable features for the pulmonary delivery of drugs due to their tolerance in the pulmonary tract, reduced absorption, limiting aggregation and adhesion [85,94]. Liposome-encapsulated drugs (mainly antibiotics) are intensively explored for lung targeting due to their improved drug pharmacokinetics and biodistribution, decreased toxicity, ability to overcome bacterial drug resistance and targeted selectivity. Several examples of commonly employed liposome-based phospholipids for antibiotic delivery are presented in Table 6.

As presented in Table 6, several liposome-encapsulated drugs are already in clinical trials, with promising results. Arikace® is an example of a drug that, according to the clinical trial database, has passed phase II. Using the data obtained [101] from two-phase 1b/2a studies (50 mg/mL liposomal amikacin formulation), in 2014 Okusanya et al. [102] analysed the dispersal of amikacin in serum, urine, and sputum samples after the administration of liposomal amikacin for inhalation at different concentrations. The results demonstrated the efficacy of a once-daily dose of 560 mg of liposomal amikacin for the treatment of CF patients with chronic infections due to *Pseudomonas aeruginosa*. 

In 2013, PulmoSphere™ (Novartis, Pharma AG, Basel, Switzerland) technology was tested in a phase I clinical trial as the component of a dry powder formulation of ciprofloxacin for inhalation in CF and non-CF bronchiectasis. The tolerability and pharmacokinetic properties of multiple-dose once-daily and twice-daily ciprofloxacin formulation were assessed in 24 patients with CF. As presented in Table 6, the formulation was well tolerated, rapidly absorbed by the lungs and into the plasma after inhalation, with low systemic exposure that suggested an absence of substantial drug accumulation [113]. However, the complex development of effective inhaled antibiotic CF treatments must consider the unique limitations of the type of infection, therapeutic agent and dose, the drug delivery system, and, most importantly, the patient [113].

Theoretically, nanoparticles hold great promise to improve drug delivery limitations in the lungs, since their surface, structure and composition can be tailored to reach the right target at the right time by providing accurate and controlled drug delivery [92,128]. Still, the success of nanotherapy depends on a number of factors, such as nanoparticle characteristics and toxicity, route of administration and physiological aspects of the lung in the presence of respiratory disease [129]. 

As mentioned above, a marked increase in mucus viscoelasticity obstructs mucociliary clearance, favours the accumulation of purulent sputum, specific to CF patients, and hinders the penetration of topically administered therapeutics either on the market or under clinical development. Engineering particles that penetrate this barrier is a critical challenge in the development of improved therapeutics for CF. Suk et al. [116] proposed the use of muco-inert fluorescent carboxyl-modified polystyrene nanoparticles covalently coated with low molecular weight diamine PEG, as transport facilitators across CF sputum. The results obtained in fresh, undiluted CF sputum demonstrated that 200 nm particles with a dense surface coating of non-mucoadhesive PEG polymers can penetrate this CF sputum barrier by accessing pores that are filled with a low-viscosity fluid and be successfully used as drug delivery systems for CF and other pulmonary diseases [116].

One promising and innovative strategy is to develop a treatment based on gene therapy, where both viral and nonviral delivery systems were investigated, and nano-selective and sustained delivery of proteasome inhibitor drugs, one of the future trends in nanomedicine. Particularly in the CF treatment region, nanoparticulate gene delivery vectors are considered a promising option due to their non-immunogenicity and the feasibility of repetitive doses, but still show low efficiency for the time being [130].

The ideal properties of nanomaterials used in gene therapy, according to a recent review of Wong et al. [131], are: nontoxicity, biodegradability, biocompatibility, stability in biological fluids, non-immunogenicity, personalized size for cellular barriers penetration (1‒100 nm), nuclease breakdown protection, high transfection efficiency, induced sustained gene expression and cost-efficiency. Although different multifunctional gene vectors have been developed in recent years, there are several practical challenges to successful gene delivery, such as nucleic acid delivery into cells, the possibility of conjugating molecules at an intracellular level, and safety and efficacy concerns. Also, the penetration of gene carriers into the highly adhesive and hyperviscoelastic mucus gel in the airways of CF patients encourages further development.

Suk et al. [127] promoted the idea of uniform expression of functional genes throughout the airway epithelium for efficient gene therapy. Therefore, the authors developed a polymeric gene carrier that compacts full-length plasmid DNA into small and highly dense PEG-coated nanoparticles. As demonstrated in 2009 [116], the engineered gene delivery platforms penetrated the viscoelastic human airway mucus gel ex vivo and mouse airway mucus ex situ by intranasal administration, without causing acute inflammation or toxicity. The ability to effectively minimize mucoadhesion of DNA nanoparticles is based on the dense surface of PEG coverage.

The most recent study on the *CFTR* gene complexed with a cationic liposome (GL67A- pGM169: Table 6) in CF epithelial cells established the first proof-of-concept that gene therapy can stabilize the progression of CF lung disease in a Phase IIb trial. Although encouraging results were obtained from the monthly administration of GL67A/pGM169 formulation in 116 patients, the treatment effect was limited and a crucial challenge nowadays is the need for more efficient vectors to deliver the DNA into the airway cells [108,132].

Another therapeutic CF challenge is the selective and sustained delivery of small molecules, namely the use of the nano-based selective proteasome inhibitor drug bortezomib with controlled and sustained drug delivery at lower doses [133] but with potential risks. In this context, the design of novel nano-based biodegradable therapeutic vehicles for CF pathophysiology without any side effects remains the main challenge for the coming years.

Overall, effective CF management should include a combination of pharmacologic treatment based on new drug developments including CFTR protein modulators, personalised biomarkers, gene therapy and nanoparticulate drug delivery along with psychological interventions in order to create hope for treatments of the CFTR defect. Also, in addition to the multiple therapeutic approaches, future treatment of the basic defect in CF must combine relevant precision medicine models, as assessed by Cholon and Gentzsch [134], in order to offer optimal and sustained treatments to all CF patients.

### 5.3. Current Practices in CF-Related Pain

Traditional pharmacotherapy for CF pain includes the use of analgesics: acetaminophen, nonsteroidal anti-inflammatory drugs (NSAID), aspirin, antispasmodic treatments and opioids.

Sermet-Gaudelus et al. [13] found that self-medication for pain is a common practice among CF patients and almost 40% of children and 50% of adults used over-the-counter analgesics.

Acetaminophen was most frequently used (59%), followed by other NSAIDs (10%) and aspirin (5%). Abdominal pain was mainly treated with antispasmodic drugs and by increased dosages of pancreatic enzymes. Up to 60% of CF children and adults reported improvement after treatment, whereas 25% the children and 10% of adults did not use any medication [13].

Although NSAIDs are a relatively good choice for pain management for CF patients during remission from chronic lung infection, their usage during the exacerbation period with intravenous aminoglycoside treatment can increase the nephrotoxicity risk. However, long-term use of NSAIDs may be unacceptable [135] due to the high risk of gastrointestinal haemorrhage. Opioids are also not an ideal alternative for refractory pain treatment in the CF population for many reasons, as they can cause constipation and thus increase the risk of distal intestinal obstruction syndrome, and can lead to cough suppression, which could reduce airway clearance results [22].

Physiotherapy and massage are also used to treat pain. Lee et al., in a study of 105 patients with CF, of whom 90% reported chronic pain, reported the positive clinical impact of a single musculoskeletal treatment session using massage/soft tissue therapy and mobilization techniques in alleviating this pain [24].

In recent years, homeopathic products and non-pharmacological remedies such as acupuncture, physical activity, rest, heat or cold, distracting activities, yoga, meditation, self-hypnosis or osteopathy have gained attention as methods to relieve pain [23,26]. In 2016, McNamara et al. [136] conducted a phase II study on the treatment of anxiety/depression and pain through relaxation yoga for patients with CF. The clinical trial enrolled 20 participants, 12 females and eight males between the ages of seven and 20, who participated in six one-on-one sessions over a 10-week period with a certified instructor who designed each yoga program specifically for the patient with CF. The results demonstrated a significant decrease in immediate anxiety and in reported joint pain from before and just after yoga therapy sessions. Therefore, yoga is safe and well tolerated by CF children and adolescents who are experiencing pain or anxiety related to their health, school, or social activities and can act as a valuable complementary therapy.

### 5.4. Nanotechnology Potential in CF-Related Pain Management

One of the primary objectives of a clinician working with CF patients should be effective pain relief. An analysis of the literature data using “nanotechnology in CF pain” as keywords on the ScienceDirect search engine, found 76 reviews and research articles related to the nanotechnology approach to CF treatment. Although this search may not be entirely representative, to our knowledge there are no data available on the use of nanomaterials in CF pain relief. Instead, a search for “nanotechnology in pain” produced about 7975 publications from 1996. According to a review article from 2018 by Beiranvand and Sorori [137], the proper treatment of pain is still a major medical challenge; therefore, different approaches using various nano-formulated materials have been employed in the last decade. According to Moradkhani et al. [138], analgesic drug delivery nanosystems have been applied in pain therapy due to their remarkable properties: enhanced drug delivery profile, increased drug action and bioavailability, targeted/sustained or prolonged drug release profile, stability in biological fluids, nontoxicity, carried drug protection up to the target cell population, and reduced side effects of the incorporated analgesic drugs [139].

Table 7 gives a brief overview of several current strategies used in pain therapy with the help of nanotechnology science. Different formulations such as liposomes, nanoparticles, nanoplates, nanocapsules, nanofibers, nanotubes, micelles and dendrimers were developed for delivering analgesics, local anaesthetics, NSAIDs or opioid compounds intended for pain therapies [140].

By analysing Table 6 and Table 7, it can be observed that various painkillers have been loaded onto different nano-formulations with similar composition to the ones for CF treatment, such as PEG, PLGA, lipids and PVA. However, what if we can design a formulation for co-administration of two therapeutic agents? Can this serve as an effective therapeutic strategy in the clinical management of CF? It has been demonstrated that the co-therapy strategy has significant promise in the clinical management of ovarian cancer, as reported by Ganta and Amiji [150]. A combination of Paclitaxel and curcumin therapy, administered as nanoemulsion formulations, was very effective in enhancing the cytotoxicity in wild-type and resistant cells by promoting the apoptotic response. With the aid of new and potentially very powerful technologies, nanomedicine in CF pain will establish distinct protocols derived from the available results for targeted and efficient disease care. Further insights into pain therapy must be gained in order to understand the underlying mechanisms that could improve CF treatment.

It is well known that problems in pain management include economic load, subjective scale of pain measurement, side effects of opioids, drug interaction, sedation and gastrointestinal problems. In this context, the progress of nanotechnology-based tools has led to numerous benefits over traditional treatment, as briefly listed below:

- biocompatibility—that is, the ability to integrate within the body without causing negative side effects or immune response;

- slow, sustained and/or controlled drug release capability, the so-called “controlled-release reservoir,” which safely delivers therapeutic agents to specific sites or cells [151] without collateral damage;

- enhanced permeability and retention effect;

- cross-cellular membrane potential and reticuloendothelial system uptake prevention owing to the nanoparticles’ small size;

- personalized surface coverage to minimize medication adherence and improve barrier penetration;

- bioavailability, to boost the delivery of therapeutic agents; 

- improved therapeutic efficacy while decreasing the intensity of side effects on healthy tissues, followed by assessment of its efficiency due to the possibility of integrating both diagnostic and therapeutic agents into a single nanovehicle;

- improved loading capacity due to high surface area;

-cost-effectiveness translated into economic impact, observed from different angles: reduced required dosages; improved shelf life, thermostability and endurance to humidity changes and potentially reduced transportation costs.

Nevertheless, nanotechnology has received significant attention because of its capacity to revolutionize medicine. Despite the impact of and progress in nanotechnology, the discussion of the potential effects on humans has just begun. Since it is widely predicted that nanotechnology will be applied in medical sciences over the next couple of years, we must mention an important challenge to researchers, namely the need to secure approval from the Food and Drug Administration (FDA) before its use in the human body [152].

## 6. Conclusions

Pain is an important complication in CF patients since inappropriate management will lead to increased pulmonary exacerbations, a decrease in treatment compliance, lower quality of life and even death. Earlier diagnosis through screening, symptom-directed treatment, effective and intensive multi-disciplinary care and vigilance could lead to a dramatic increase in the quality of life and life expectancy of CF patients. Therefore, correct pain assessment should be included in the algorithm for the evaluation and monitoring of disease progression in patients with CF, as an indicator of the unfavourable, rapidly progressing and severe evolution of the disease. Currently, pain is underestimated and no guidelines and tools have been developed to identify its presence and severity in the clinic. Also, there are no recommendations for pain management in both children and adults with CF, although pain relief may increase compliance to regular treatment of the disease. A particular focus of this review article is the nanotechnology applicable in CF treatment, with a special emphasis on pain. Even though the available data demonstrate that nanotechnology has great potential for revolutionizing CF and its associated pain, we believe that a multidisciplinary approach is needed for optimal care of CF patients. Overall, pain control should be a priority in the complex multidisciplinary therapeutic protocol of CF—essential to prevent complications and unfavourable evolution, and thus to increase quality of life.

## Figures and Tables

**Table 1 jcm-08-01023-t001:** Pain regions and prevalence in children/adolescents and adults.

Pain location	Prevalence in Children/Adolescents	Prevalence in Adults
Abdominal pain	42–100% [11,14,13,23,30,31,32]	19–50% [13,20,27,33,34,35]
Gastrointestinal pain	10% (stomach) [13]	10–51% [13]
Chest pain	10–38% [11,13,14,23,32]	9–72% [12,13,20,26,27,33,35]
Back pain (mid-back, lower back and/or both)	6–16% [13,32]	19–70% [12,13,20,27,33]
Head and neck pain	13–42% [11,14,23,32]	6–64% [12,13,20,27,33,34,35]
Cervical	3% [13]	10–28% [13,34]
Limbs (upper and/or lower limp pain)	11–19% [13,32]	6–27% [12,20,27,35]
Musculoskeletal pain	3–19% [13,32]	6–44% [13,34,35]

**Table 2 jcm-08-01023-t002:** Gastrointestinal manifestations in CF patients.

Organ	Complication		References
Pancreas	Exocrine pancreatic insufficiency; Pancreatitis;	Chronic pancreatitis; Pancreatic lithiasis; Malignancy;	[38]
Gastrointestinal tract	Meconium ileus, distal intestinal obstruction syndrome; Rectal prolapse; Constipation; Atresia; Intussusception; Intestinal invagination; Appendiceal abscess;	Gastroesophageal reflux disease; Gastritis, gastroduodenal ulcer; Fibrotic colonopathy; Dysbiosis; Inflammation; Malignancy;	[39,40,41,42]
Liver	Prolonged physiological jaundice; Multilobular cirrhosis; Portal hypertension; Splenomegaly; Hypersplenism;	Neonatal hepatitis; Hepatic steatosis; Liver enzyme abnormalities; Focal bile fibrosis;	[43,44]
Gallbladder	Multilobular biliary cirrhosis; Gallbladder abnormalities;	Gallbladder lithiasis, cholecystitis; Malignancies.	

**Table 3 jcm-08-01023-t003:** Topography and origin of abdominal pain.

Pain Localization	Embryonic Origin	Anatomic Localization	References
Epigastrium	Foregut	Proximal to the Treitz ligament, including the hepatobiliary system and the spleen	[51]
Periumbilical	Midgut	Treitz ligament to the colon’s hepatic angle	
Hypogastrium	Hindgut	Colonic liver angle up to rectum	

**Table 4 jcm-08-01023-t004:** Relationship between abdominal pain localization and aetiology.

Pain Localization	Aetiology	References
Epigastrium	Gastroesophageal reflux disease, gastritis, ulcer, pancreatitis	[52,53]
Periumbilical	Gastroenteritis, intestinal obstruction syndrome, invagination, appendicitis	
Right hippocampus	Hepatobiliary disorder, pancreatitis and pneumonia	
Left hippocampus	Gastritis, pancreatitis, splenic infarction, pneumonia	
Right iliac fossa	Distal intestinal obstruction syndrome, appendicitis, renal or ovarian disorder	
Left iliac fossa	Constipation with faecal impaction, kidney or ovarian disease	
Diffuse abdominal pain	Gastroenteritis, intestinal obstruction syndrome, peritonitis.	

**Table 5 jcm-08-01023-t005:** Pulmonary manifestations in patients with CF.

Organ	Complication	References
Pulmonary parenchyma	Chronic pulmonary infections (pneumonia, bronchopneumonia); Obstruction of the airways; Haemoptysis; Bronchiectasis; Pulmonary atelectasis; Pulmonary fibrosis; Chronic respiratory failure;	[59,61]
Pleura	Pleurisy; Pachypleuritis; Pneumothorax; Haemothorax;	
Sinus	Sinusitis.	

**Table 6 jcm-08-01023-t006:** Nanomaterials under development for CF treatment.

Formulation Type	Materials + Drug	Key Summary	References
Liposome- encapsulated drugs (aqueous dispersions)	DSPC/DMPG + Tobramycin;	Significant increase of drug residence time within rat lungs infected with *Pseudomonas aeruginosa.*	[95]
	DPPC/DMPG + Tobramycin;	Efficient bactericidal activity on chronic pulmonary infection caused by mucoid *Pseudomonas aeruginosa.*	[96]
	DPPC/DMPG + Gentamycin+Ga (III) nitrate	Optimized and efficient co-delivery, reduced Ga toxicity, antimicrobial activity and complete eradication of antibiotic-resistant clinical isolates of *Pseudomonas aeruginosa.*	[97]
	DSPC/Chol + Tobramycin + bismuth-ethanedithiol;	Non-toxic and stable formulations that penetrate sputum, reduce quorum sensing molecule and virulence factors production and inhibit the growth of biofilm-forming clinical strains of *Pseudomonas aeruginosa*.	[98,99]
	DPPC/Chol + Amikacin	Sustained and targeted release, biofilm and infected *Pseudomonas aeruginosa* mucus penetration, superior efficacy on both in vivo and phase II clinical trials of inhaled liposomal amikacin.	[100,101,102]
	DC-Chol/DOPE + CFTR cDNA	In vivo administration of CFTR cDNA transfection of cationic liposomes corrected the ion transport defect in the airways of the mouse model of CF created by insertional mutagenesis.	[103]
	DOTAP+ pCMV-CFTR expression vector	Pilot randomised, double-blinded study of cationic liposome complex single-dose administration to the nasal epithelium of eight CF patients; significant gene transfection, no adverse effect and no evidence of inflammation.	[104]
	DODAG/DOPE/PEG4600-Chol + Plasmid DNA encoding luciferase	New cationic liposome systems mediated efficient transfection of healthy murine lung tissue in vivo without significant inflammation; the obtained carriers could form the basis for nucleic acid therapeutic strategies for CF gene therapy.	[105]
	GL67A/pGM169 (GL67/DOPE/DMPE-PEG5000 + Plasmid DNA)	Novel cationic lipid-PEG formulation (GL67A) associated with plasmid DNA (pGM169); reduced inflammation and sustained pulmonary gene expression after in vivo aerosol delivery; single-dose phase I and IIa safety and gene expression study of pGM169 or GL67A administered to the nose and lungs on CF patients (ClinicalTrials.gov number: NCT00789867); randomised, double-blind, placebo-controlled, phase IIb clinical trial (ClinicalTrials.gov number NCT01621867).	[106,107,108,109,110]
Solid lipid particles (saline solutions or dry powders)	Stearic acid/PC+ Myriocin	Good uptake and delivery of myriocin-loaded nanocarrier, significant reduction of lung infection and reduced inflammation on CF mice.	[111]
	Chol/lecithin + Tobramycin	Formulation and in vitro evaluation: large surface area, low bulk density, good flowability, size and shape suitable for use in carrier-free dry powder inhalers.Pilot study on CF patients: high lung deposition and reduced systemic bioavailability determined by a pharmacoscintigraphic method.	[112,113]
	DSPC + Tobramycin	PulmoSphere™ porous particles are obtained by spray-drying method in four steps: emulsion-based feedstock preparation by high-pressure homogenization of perfluorooctyl bromide, DSPC, tobramycin and calcium chloride in water; atomization with a twin fluid nozzle into a hot air stream; drying of the emulsion droplets and collection of resulting dry powder comprising porous spheroidal particles.	[114]
	DSPC + Ciprofloxacin	Phase I, randomized, single-blind, placebo-controlled, dose-escalation study in patients with CF; the formulation was well tolerated, has targeted and sustained release, minimal systemic exposure and no apparent accumulation of ciprofloxacin over seven-day treatment period.	[115]
Polymeric nanoparticles (aqueous dispersion or dry powders)	PS-PEG	200 and 500 nm nanoparticles; the dense surface coating of the 200 nm non-mucoadhesive nanoparticles helped the penetration of CF sputum.	[116]
	PSA-PEG	Biodegradable nanoparticles with an average hydrodynamic diameter of 173 nm, prepared using a conventional solvent diffusion method rapidly penetrated sputum expectorated from the lungs of patients with CF due to a dense surface coating of low PEG.	[117]
	PVA-Alg/PLGA + Tobramycin and CS-Alg/PLGA + Tobramycin	2 types of PLGA nanoparticles were prepared by a modified emulsion/solvent diffusion technique for the production of dry powders for antibiotic inhalation; the formulations displayed good in vitro antimicrobial activity against *Pseudomonas aeruginosa* planktonic cells, and differentiated in vivo biodistribution and deposition pattern, dependent on the nanoparticle composition.	[118]
	PLGA + Pirfenidone	Intratracheal administration of biodegradable pirfenidone nanoparticles in bleomycin-induced pulmonary fibrosis in mice, determined sustained lung delivery and anti-fibrotic enhanced efficacy.	[119]
	PLGA-PEG + PS-341	Mono-dispersed and spherical in shape loaded nanoparticles were synthesized using non-polar core of oil-in-water microemulsion technique with PEGylated phospholipid DSPE-mPEG^2000^ as the emulsifier; the drug delivery system provided controlled and sustained PS-341 delivery for selective inhibition of proteostasis.	[120]
	Transferrin-gelatine/chloroquine/calcium+ DNA	A plasmid DNA encoding CFTR and gelatine nanoparticle coacervate transfected in the presence of calcium and transferrin, resulted in CFTR expression in over 50% of the cells; effective transport activity.	[121]
	CS/FAP-B + DNA encoding luciferase	250 nm nanoparticles nebulized to mice lungs determined 16-fold increase of gene expression compared with CS-DNA NPs without FAP-B receptors.	[122]
	PEI+ miRNA and CS + miRNA	Non-toxic miRNA-PEI nanoparticles (300 nm) facilitated greater uptake into CFBE41o- cells, and efficient delivery than miRNA-CS nanoparticles (115 nm).	[123]
	PEI + HA + plasmid DNA encoding luciferase	Inhalable dry microparticle form of mannitol, encapsulating ternary complex composed of plasmid DNA, disulphide-crosslinked low molecular weight linear PEI and HA as a gene carrier, improved CF artificial sputum penetration and transport but with gene transfer agents aggregation.	[124]
	PEI + DNA encoding luciferase	Aerosol delivery complex with plasmid DNA and branched PEI repeatedly administered to airways of mice showed no detectable toxicity being suitable for treatment of chronic lung diseases.	[125]
	Poly-l-lysine-PEG + DNA encoding CFTR	Rod-shaped DNA nanoparticles with different PEG molecular weights (2, 5 or 10 kDa) provided partial protection against DNase I digestion and exhibited the highest gene transfer to lung airways following inhalation in BALB/c mice, but were immobilized in freshly expectorated human CF sputum due to inadequate PEG surface coverage.	[126]
	PEI-PEG + DNA encoding CFTR	Synthetic gene carrier platform composed of PEG (5 kDa) and branched PEI (25 kDa) penetrated human CF mucus due to highly dense PEG coating, achieved uniform airway distribution and prolonged lung retention, and enhanced gene transfer to mouse lung with no inflammatory responses.	[127]

**Table 7 jcm-08-01023-t007:** Nanomaterials for pain therapy.

Formulation Type	Materials + Drug	Key Summary	References
Liposomes	PEG + Methylprednisolone hemisuccinate;	80 nm sterically stabilized drug loaded nanoliposomes were used to treat Lewis rats with adjuvant-induced arthritis	[141]
	Shea butter lipid nanoparticles + Nimesulide	90 nm polydisperse loaded lipid nanoparticles presented significant in vivo antinociceptive activity compared with free nimesulide	[142]
	Liposomes + Celecoxib+ embedded in hyaluronic acid gel	Celecoxib loaded liposomes showed high efficiency in pain control and cartilage protection on in a rabbit knee osteoarthritis model after intra-articular injection	[143]
	Lipids+ Bupivacaine	Single dose of the liposomal formulation reduced the pain over 72 h and decreased opioid requirements in 184 patients undergoing haemorrhoidectomy	[144]
Nanoparticles	Anti-ICAM-1 (Intercellular Adhesion Molecule 1) + Loperamide HCl	Administration of targeted nanoparticles exerted analgesic and anti-inflammatory effects in peripheral painful inflamed tissue on adult male Wistar rats	[145]
	Poly(amidoamine) (PAMAM) dendrimer + esterase activated morphine prodrugs	Esterase-sensitive prodrugs administration enhanced the sustained release of morphine, which extended the action of morphine-induced analgesia in an animal pain model from 2 to 6 h	[146]
	Butylcyanoacrylate nanoparticles + polysorbate 80+Endomorphin-1	Intravenously administered nanoparticles act as an analgesic agent to target the brain	[147]
Nanofibers	PLGA nanofibers + Lidocaine	The nanofibers introduced into the epidural space of rats after laminectomy provided a sustained release of lidocaine for more than two weeks	[148]
	PVA + Meloxicam	Nanofiber mats loaded with meloxicam as a transdermal analgesic drug delivery system	[149]

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
