# Peer review of "Current Practices and Potential Nanotechnology Perspectives for Pain Related to Cystic Fibrosis"

_jcm, 2019, doi:10.3390/jcm8071023_

Round 1

Reviewer 1 Report

The manuscript presented by Trandafir et al is a well written review giving a broad insight on pain affecting CF patients, a key aspect which critically worsens patients’ quality of life, but is often underestimated.

For the scientific relevance of the subject matter and the quality of the manuscript itself we recommend publication in this journal. However, some major modifications are needed.

In particular, I suggest the authors to be more focused on the proposed topic. Indeed, the paper has to be dramatically shortened to increase readability.

Major comments:

Subsection 2 - “Etiopathogenic aspects”: The paragraph should be heavily shortened. I also suggest to move this subchapter after the description of “Pain and CF” in order to provide a mechanistic explanation to chronic widespread pain or organ-specific pain. Generally speaking, the link between inflammation and pain should be highlighted and better explained.

Subsection 3 - “Histopathological aspects” is a bit prolix and not fully necessary to the purpose of the review. I suggest to include only key information in the following “Pain in CF” chapter, so that information on organ-specific pathological aspects is reported in the relevant context.

Subsection 4 – “Pain in CF”: Instead of listing methods that are generally used to score pain, I suggest to explain the methods that are actually and specifically used in CF Example: Medical Chart Review, Multidimensional Pain Inventory etc… 

Subsection 6.1: Starting from line 561 up to line 683, the review loses completely the focus on CF-related pain. After a small introduction to investigational therapies targeting CF basic defect in Section 6.1, in Subsection 6.2 the authors describe nanotechnologies applied to drugs to treat CF. However, no reference to pain therapeutic options is provided. Albeit this complete insight on the new approaches for CF therapy is extremely useful and interesting, this is not in line with the focus of the review. The author should amend the manuscript accordingly, or should explain how these nanotechnologies could be exploited for pain management. Are there any new approaches in the nanotechnology field to target pain that can be used in CF patients? Also the conclusions section does not include any consideration on nanotechnologies on pain in CF. I suggest no to include reference to nanotechnologies in CF pain treatment unless more evidence is provided. The title of the presented review should be modified according to the content of the manuscript.

Minor comments:

98 “encourages” , final “s” has to be removed.

112 “Even if, neutrophils represent”  “,” has to be removed.

115 “CFTR gene”: when you are referring to the gene, CFTR has to be written in italic.

281 An appropriate reference is missing.

345 “The prevalence of 59%, management, and features of pain in a population of lung transplant 345 candidates were determined in 2015 by Michel-Cherqui et al.” Sentence should be rephrased.

685 “Pain is an important complication in CF patients since appropriate treatment will lead to increased pulmonary exacerbations, decrease in treatment compliance, and thus, lower quality of life and death”. Sentence should be rephrased.

Author Response

On behalf of the manuscript authors, I would like to sincerely thank you for your kind answer

and valuable advices given to improve this review paper. All your comments are fare and

justified considering the body of the text. All the modifications were highlighted in red.The

answers to the reviewer comments are listed in the attached document.

Best regards,

Gianina Dodi

Reviewer 2 Report

The current manuscript seems to be a detailed account of the current practices in cystic fibrosis related etiology and pain. However, the nanotechnology based perspective is not discussed in details.

Although, this manuscript is a fine review, it will be significantly improve if the authors could discuss the following in a detail:

Current practices in CF management using nanotechnology

Difficulties and challenges in delivering nanoparticle based medications in already inflamed/ compromised lungs in patients

How the nanoparticle based interventions might offer better solutions in terms of controlled release of drugs/ therapeutics

A comparative analysis of traditional versus nanoparticle based interventions.

Discuss the existing/ future  genetic interventions where nanotechnology based delivery methods could benefit.

These suggestions will definitely improve this review by exploring the nanotechnology based therapeutics/ pain management. 

Author Response

On behalf of the manuscript authors, I would like to sincerely thank you for your kind answer and valuable advices given to improve this review paper. All your comments are fare and justified considering the body of the text. All the modifications were highlighted in red. The answers to the reviewer comments are listed in the attached document.

Best regards,

Gianina Dodi

Round 2

Reviewer 1 Report

The manuscript has been properly revised according to the reviewer suggestions and is now more readable, fluent and focused on the main theme. The authors now better focus on pain in CF and  provide a perspective on nano-based drugs to target CF-related pain. 

Only some minor typos still have to be fixed:

Line 164: Revise as follows: "In another study, 250 adults with CF were evaluated in a complex longitudinal study that explored the..."

Line 267: "Prenatal and postnatal changes also interest the liver and the bile ducts": replace "interest" with "affect".

Line 785 "Even though,..." remove comma.

Author Response

On behalf of the manuscript authors, I would like to sincerely thank you for your valuable instructions given to improve this review paper. All the modifications were highlighted in red. The answers to the reviewer comments are listed below, as follows:

Comments and Suggestions of Reviewer 1-Round 2 for Authors

The manuscript has been properly revised according to the reviewer suggestions and is now more readable, fluent and focused on the main theme. The authors now better focus on pain in CF and provide a perspective on nano-based drugs to target CF-related pain. 

Response: The authors appreciate all your suggestions. 

Only some minor typos still have to be fixed:

Point 1: 

Line 164: Revise as follows: "In another study, 250 adults with CF were evaluated in a complex longitudinal study that explored the..."

Response 1: The sentence was revised according to the reviewer comment.

Point 2: 

Line 267: "Prenatal and postnatal changes also interest the liver and the bile ducts": replace "interest" with "affect".

Response 2: The word "interest" was replaced with the word "affect" according to the reviewer comment.

Point 3: 

Line 785 "Even though,..." remove comma.

Response 3: Comma was removed.